# Structural and Physiological Traits of Compound Leaves of *Ceratonia siliqua* Trees Grown in Urban and Suburban Ambient Conditions

**DOI:** 10.3390/plants12030514

**Published:** 2023-01-23

**Authors:** Sophia Papadopoulou, Aikaterina L. Stefi, Maria-Sonia Meletiou-Christou, Nikolaos S. Christodoulakis, Dimitrios Gkikas, Sophia Rhizopoulou

**Affiliations:** Section of Botany, Department of Botany, Faculty of Biology, National and Kapodistrian University of Athens, 15784 Athens, Greece

**Keywords:** anatomical traits, chlorophylls, phenolics, pollution, specific leaf area, water deficiency

## Abstract

*Ceratonia siliqua* L. (carob tree) is an endemic plant to the eastern Mediterranean region. In the present study, anatomical and physiological traits of successively grown compound leaves (i.e., the first, third, fifth and seventh leaves) of *C. siliqua* were investigated in an attempt to evaluate their growth under urban and suburban environmental conditions. Chlorophyll and phenolic content, as well as the specific leaf area of the compound leaves were determined. Structural traits of leaflets (i.e., thickness of palisade and spongy parenchyma, abaxial and adaxial epidermis, as well as abaxial and adaxial periclinal wall) were also investigated in expanding and fully expanded leaflets. Fully expanded leaflets from urban sites exhibited increased thickness of the lamina and the palisade parenchyma, while the thickness of the spongy parenchyma was thicker in suburban specimens. The palisade tissue was less extended than the spongy tissue in expanding leaflets, while the opposite held true for the expanded leaflets. Moreover, the thickness of the adaxial and the abaxial epidermises, as well as the adaxial and abaxial periclinal wall were higher in suburban leaflets. The chlorophyll content increased concomitantly with the specific leaf area (SLA) of both expanding and expanded leaflets, and strong positive correlations were detected, while the phenolic content declined with the increased SLA of expanding and expanded leaflets. It is noteworthy that the SLA of expanding leaflets in the suburban site was comparable to the SLA of expanded leaflets experiencing air pollution in urban sites; the size and the mass of leaf blades of *C. siliqua* possess adaptive features to air pollution. These results, linked to the functional structure of expanding and expanded successive foliar tissues, provide valuable assessment information coordinated with an adaptive process and yield of carob trees exposed to the considered ambient conditions, which have not hitherto been published.

## 1. Introduction

*Ceratonia siliqua* L. (carob tree) is a slowly growing, evergreen, sclerophyll, dioecious species, which is widespread as a native plant in the Mediterranean Basin, although thousands of years of trade by people in the region has meant that this species has become widely naturalized [1,2]. *C. siliqua* is considered a phylogenetically primitive species of tropical origin that has been cultivated in the Mediterranean area since historic times [3,4,5,6], and is an economically important plant [7,8,9,10,11]. Additionally, it has been used for afforestation in semi-arid regions [12,13,14]. In the eastern Mediterranean, *C*. *siliqua* is subjected to a prolonged drought period, which follows the main leaf growth period of the evergreen sclerophyll shrubs during spring [15,16,17,18]. 

The pinnate, compound leaves of *C. siliqua* expand within a 4-month period, that is, from March to June; then their expansion is ceased. However, they are maintained on the trees, exposed to the ambient environmental conditions for approximately the oncoming 20 months [18,19,20,21,22]. The root apices of the deep tap roots of *C. siliqua* sustain water flux to the leaves during the prolonged periods of topsoil drying [23,24,25,26].

Urban greenery plays an important role in improving the quality of the urban environment. However, air pollutants cause a variety of adverse effects and damage to the leaves of plant species. During the past four decades, there have been several studies of plant responses to air pollution. It has been published that different plant species elicit the environmental quality in which they grow by changing their leaf anatomical and physiological properties; besides morphological, physiological and anatomical features, biosynthesis of phenolic compounds, even as metabolically terminal products, has been considered to be a common response of Mediterranean plants, directly related to the environmental stresses [27,28,29,30,31,32]. 

The air of the industrialized Mediterranean counties and metropolitan cities contains an array of pollutants which are potentially toxic to plants; usually, the growth of leaves exposed to air pollutants has been inhibited [33,34,35,36], affecting plant productivity. However, studies linked to the response of plant species to air pollution provide some evidence for differences in sensitivity among successive leaves [37,38,39]. Additionally, vegetation can alter the dispersion of pollutants from roads into parks and stands of trees may change exposure to air pollutants [40]. Plants from polluted sites possess important phenotypical alterations, especially regarding their colors, shapes, leaf length, width and area, and petiole length. As leaves represent the main surfaces of plant canopies, where energy and gases are exchanged, they are the most sensitive parts to be affected by air pollution; therefore, leaves may serve as sensors of air pollutants, at various stages of their development, indicating that plants do survive in polluted environments [41]. The benefit of using plants as biosensors is their uncomplicated deployment in the field. Moreover, monitoring based on biosensors is cheap compared to the costly physico-chemical monitoring methods [42,43].

*C. siliqua* is being investigated as a potential bio-monitor plant for urban habitats, growing in industrialized, urban and suburban areas, exhibiting morphogenetic plasticity and tolerance to drought stress conditions [38,44,45,46]. This evergreen and long-lived species, which does not require any irrigation and cultivation treatments, is widely planted along urban street corridors in the Athens Metropolitan area (Greece) [14,37].

Since the problems caused by the long-term exposure to air pollution are potentially toxic to plants, studying the characteristics of plant life in urban areas is essential and will improve human life in polluted atmospheres [47,48,49]. In addition, the effect of air pollutants on plant growth depends not only on the concentrations of air pollutants, but also on the stage of leaf development; even though there is information on characteristics of Mediterranean plants [17,33,41,43], it should be noted that detailed data are far from definitive. In considering the above-mentioned issues and *C. siliqua* as one of the most distinct and resistant evergreen phanerophytes thriving at the dry and arid edge of the Mediterranean region, we launched this investigation in order to evaluate the effect of air pollution on structural and physiological traits of young, expanding, and fully expanded successively developed leaves, which according to the best of our knowledge have not hitherto been published. 

## 2. Results

### 2.1. Leaf Histology

In Figure 1a, a leaf cross-section from the suburban KE research site (Figure 2, Table 1) harvested from trees grown on the mountain Hymettus is depicted. In the cross-sections of successively expanded leaves of *C. siliqua* (Figure 1b–e), harvested from urban sites (Figure 2, Table 1), it is shown that large amounts of phenolics accumulated in the epidermal and the mesophyll cells (Figure 1b–e). 

In Figure 1b–e, cross-sections of the third expanded leaflet (see Section 4.1) from the four selected urban sites are presented. The leaflet was well-preserved and the major xeromorphic features (thick cuticle on the epidermal cells, thick outer periclinal wall of the epidermal cells, compact and well-developed palisade, accumulation of secondary metabolites, stomata only on the abaxial epidermis, that is, hypostomatous leaflets) were observed (Figure 1). It is noteworthy that at all developmental stages, the leaves collected from the urban trees were always thicker, in comparison with the leaves collected from the suburban site (Table 2); also, the samples from trees grown in the NG site seemed to be a bit more xeromorphic (Figure 1e). 

The adaxial epidermis of leaflets was approximately from 2- to 3-fold thicker than the abaxial epidermis (Table 2). The smaller thickness of both the adaxial and the abaxial epidermis was detected in leaflets collected from the polluted site NG, while higher values of thickness of the abaxial epidermis (where stomata are present) were observed in leaflets collected from the suburban site KE (Table 2).

The thickness of the palisade tissue was thinner than that of the spongy tissue in expanding leaflets from both urban and suburban sites (Table 2). In expanded leaflets from urban sites, the thickness of the palisade tissue was thicker than that of the spongy tissue, while the opposite held true for samples from the suburban site (Table 2). The thickness of the palisade tissue of the seventh expanding leaflet was almost the same in all the polluted urban sites, and thicker than that of the suburban site. Additionally, the thickness of the palisade tissue of the fully expanded leaflets (12-months old) was thinner in the suburban site in comparison with that the urban sites (Table 2). In successively expanding leaflets of *C. siliqua* in urban sites, the palisade tissue increased concomitantly with the spongy tissue, while in expanded leaflets, the palisade tissue occupied more than half of the mesophyll. Thus, the thickness of palisade and spongy parenchyma of the third expanding leaflets were positively correlated (y = 0.7057x + 124.75, *r* = 0.7918, *p* < 0.05), while they were negatively correlated (y = −0.2783x + 440.62, *r* = 0.9933, *p* < 0.05) in the third expanded leaflets. Additionally, the thickness of both the palisade and spongy parenchyma of leaflets collected from the suburban site was sustained, rather constant (mean values = 263.36 μm and 317.28 μm, respectively) (Table 2); in fact, the palisade varied from approximately 247 μm (first expanding leaf) to 275 μm (seventh expanding leaflet), and the spongy one from approximately 300 μm (first expanding leaf) to 338 μm (seventh expanding leaflet). The palisade cells were impregnated with osmiophilic material, which was actually condensed phenolics. A part of the adaxial epidermis seems to be free of phenolics; this is due to an artifact. Many times, phenolics leach from the vacuole during the first stage of the tissue fixation, and destroy the membranes and thus all the organelles of the cells. 

The thickness of the periclinal wall of both the adaxial and abaxial epidermal cells was thicker in leaflets (at any developmental stage) collected from the suburban site than those from the urban sites (Table 2). Additionally, the thickness of the periclinal wall of the abaxial epidermal cells was quite similar between the fifth and seventh expanding leaflets, as well as with that of the third expanded leaflets collected from polluted sites. 

In Figure 3, cross-sections from successively expanding leaflets of *C. siliqua* from all the considered research sites are presented. In Figure 3a, under the designation using initials KE, images from leaflets of the suburban area are presented. In Figure 3b, images of cross-sections from leaflets of the trees grown in the less polluted site PF are presented. The leaflets of the heavily polluted sampling site KL (Figure 3c) appear a bit more compact with abundant phenolics. The intercellular spaces are wider in the lower irregularly shaped spongy parenchyma cells than in the palisade cells. Additionally, spongy parenchyma cells appear to be more loosely arranged into the leaflets KE5 and KE7, as well as ME5 and ME7. In Figure 3e, images of cross-sections from leaflets of the trees grown in the polluted site NG are presented. 

### 2.2. Physiological Traits

Elevated values of specific leaf area (SLA) were estimated in expanding and expanded leaves harvested from the urban sites in comparison with those of the suburban site (Table 3), which were mainly due to low dry weight.

Positive linear correlations of total chlorophyll content, of expanding (Figure 4a) and fully expanded (Figure 4b) leaves from all the considered sites in each of the studied successive leaf (first, third, fifth, seventh), with the corresponding SLA were detected (Table 4). Analytically, a significantly positive correlation was detected between total chlorophyll and SLA in the third (*r* = 0.9873), fifth (*r* = 0.9994) and seventh (*r* = 0.9915) expanded leaves (Table 4), whereas expanding leaves exhibited weaker values (Table 4). The total chlorophyll content was higher in fully expanded compound leaves when compared to the respective expanding compound leaves; also, the first expanding leaf attained slightly lower chlorophyll content than the subsequent leaflets from the third, fifth and seventh leaves. Furthermore, leaves collected from the urban sites (ME, KL, NG, PF) accumulated higher amounts of chlorophyll than those from the suburban site (KE). 

The phenolic content of expanding (Figure 4a) and fully expanded (Figure 4b) compound leaves of *C. siliqua* from the considered sites, in each of the considered successive leaves (first, third, fifth, seventh; see Section 4.1) was plotted versus the corresponding SLA (Table 3), and negative correlations were determined in specimens from all sites. Expanded leaves, especially the third expanded leaf, exhibited the highest coefficient of the negative linear regression (Table 4, *r* = 0.9663). Expanding leaves grown in suburban and urban sites contained higher amounts of phenolics than their expanded counterparts. With respect to the variables of foliar phenolic content, SLA, and structural traits of expanded leaflets, the considered specimens seem to be classified in two groups (Figure 5a); one group includes specimens from the suburban site (KE), while the other group includes those from the urban sites (ME, KL, NG and PF).

### 2.3. Principal Component Analysis

The Principal Component Analysis (PCA) (Figure 5a) shows that PC 1 and PC 2 reveal significant (99.78%) differentiation between suburban and urban leaves of *C. siliqua*. The considered traits of successively grown leaves of carob indicate a substantial grouping of the specimens collected from the carob trees from suburban and urban sites. In the biplot (Figure 5b), the vectors represent the contribution that each variable has in the two-dimensional spaces. The PCA-based biplot indicates differences and separation among the more and less polluted sampling sites presented by red spots and capital letters (see also Table 1).

## 3. Discussion

The effect of the air pollutants on plant structure and function has been the focus of interest for many investigators, while plants exhibit contrasting responses to air pollution. It has been shown that different plant species elicit the environmental quality in which they grow by changing their leaf anatomical and physiological properties to provide a reasonably accurate assessment of habitat quality [49,50,51]. Air pollutants may cause changes in stomatal opening, photosynthesis, photosynthetic pigments’ content, plant productivity and structural traits (e.g., thicker epidermal cells, longer trichomes) [52,53]. For example, leaves of urban trees of *Platanus orientalis* possess a significant decrease in stomata density, epidermis, and spongy mesophyll thickness [54], but a significant increase in cuticle and palisade thickness. Additionally, in *Rhododendron* *pulchrum* exposed to air pollution, a reduced stomatal density was found, while *Ginkgo biloba* leaves exhibited small stomatal density and elevated mesophyll thickness [55]. 

Fully expanded leaves of *C. siliqua* grown in the urban sites and exposed to air pollution possess a statistically significant increase in the thickness of their lamina and palisade mesophyll, as well as in the ratio of palisade versus spongy parenchyma, in comparison to the leaves from suburban trees, which is in agreement with results from other species [56,57,58]. Elevated values were estimated in samples of the NG site, probably because trees grown there are exposed to the air pollution emitted by continuous, diurnal traffic of cars. The leaf spongy mesophyll tissue enables carbon capture and provides mechanical stability; also, unlike many other biological tissues, which remain confluent throughout development, the spongy mesophyll develops from an initially confluent tissue into a network of cells with a relatively large proportion of intercellular airspace [59]. In addition, urban leaves possessed a significantly thicker thickness of both the adaxial and the abaxial epidermises, as well as the periclinal wall of adaxial and abaxial epidermal cells.

It is noteworthy that in the expanded leaves harvested from the suburban location, the spongy mesophyll was thicker than the palisade, while the opposite held true for leaves harvested from urban and air-polluted sites (Table 2). Additionally, the thickest thickness of the photosynthetic palisade parenchyma was estimated in expanded leaflets collected from polluted urban sites (ME, KL, NG). It has been argued that the palisade tissue may help the distribution of light more uniformly to chloroplasts within the leaf blade and may be related to the amount of collimated light within ambient environmental conditions [60,61], evidencing the complexity of factors that regulate the path of the light beam from the leaf surface into the internal structure of the photosynthetic tissue. In addition, it has been published that plants of carob do sustain a positive carbon balance during the growth season (when young leaves expand), indicating that at least some leaves must show reduced stomatal sensitivity to seasonal soil drying [23]. 

Additionally, compound leaves of *C. siliqua* exposed to air pollution appear to contain higher total chlorophyll than those harvested from suburban stands of trees, which is in agreement with earlier results from other Mediterranean plant species growing in the Athens metropolitan area, reflecting the response of plants to the quality of the air. Due to NO_x_ air pollution in urban sites, the chlorophyll concentration increased, while it decreased if the pollutant was ozone in suburban sites [51]. The opposite held true for other plant species; for example, in *Celtis occidentalis* the chlorophyll content of the leaves was lower in urban sites [50]. 

A strong positive correlation was detected between SLA and chlorophyll content, while a negative one was found between SLA and phenolic content in expanded carob leaves. It is worth mentioning that although suburban carob leaves were coming through their buds and being extended two weeks later than the leaves grown in the urban sites, their development (and growth) was completed within a shorter period of time. In this context, it became apparent that SLA values of expanding leaflets in the suburban site (KE) were comparable to those of fully expanded leaflets experiencing air pollution in urban sites, over the period of a year (Table 3). 

Plants’ adaptation to abiotic environmental stresses is associated with metabolic adjustments that lead to the accumulation of phenolics that ameliorate the damaging effects of abiotic stress [62,63,64]. Additionally, substantial phenolic accumulation can occur in nitrogen-limited environments [20,65], which are actually the natural ecosystems of carob trees with their long-lived leaves [20,66], possessing an interesting functional leaf surface area [67,68,69]. *C. siliqua* is a species that synthesizes and accumulates phenolics in the young stages of leaf development, as protective substances against biotic and abiotic stress factors [30]. Thereafter, with the increment of leaf age (as well as leaf area and leaf mass) and maturation, the phenolic content of expanded leaves declines, which may be linked to the maintenance costs of these long-lived leaf tissues [26,51]. In contrast to expanding leaves, the successively expanded leaves in each urban site contain almost the same amount of phenolics and possess comparable SLA values; thus, it is indicated that these leaves have attained their maturation phase and their growth has ceased. With respect to differences among sites, a higher SLA detected in urban sites (Table 3) compared to the suburban samples could result, on one hand, in enhancing air pollutants’ uptake, and on the other hand, an increased photosynthetic area. Phenolic content accumulated in expanding and expanded leaves collected from the suburban (KE) site is higher than those from the urban sites; it seems likely that suburban plants grown in the natural environment of the aesthetic forest, invest more in protection against environmental stress factors than urban carob plants. These results were broadly in line with other results linked to leaf total phenolic metabolites that protect against reactive oxygen species [70]. 

Given the long lifespan of the compound leaves of *C. siliqua*, the foliar traits ascribed to trees grown in suburban and urban sites of a metropolitan city provide valuable information linked to the adaptive process of primary productivity and the yield of carob trees upon exposure to the considered ambient conditions. The presented functional and anatomical traits of the compound leaves of carob help further our understanding of the development of this species in an air-polluted environment in the Mediterranean region. Additionally, they provide some evidence that carob trees exposed to long-term air pollution do produce new leaves, and pollutant-induced foliar changes may be countered by an increase in SLA.

## 4. Materials and Methods

### 4.1. Plant Material and Sampling Sites

The present study was conducted on expanding (3-month-old) compound leaves of the current year and fully expanded (12-month-old) compound leaves of the previous year, from stands of trees of *Ceratonia siliqua* L. (Leguminosae) grown in five urban and suburban ambient conditions (Figure 2b). The selection of the sampling sites was based on the existence of nearby air quality monitoring stations; therefore, four urban areas in the center of the city of Athens (ME, KL, NG, PF) and a suburban area located on the slope of the Hymettus mountain, near the campus of the National and Kapodistrian University (KE, altitude 618 m) and the aesthetic forest of Kessariani (https://www.philodassiki.org/, accessed on 7 January 2021) were selected. The four heavily polluted sites, according to the air pollution records provided by the Ministry of Environment and Energy (https://ypen.gov.gr, accessed on 1 December 2022) are located within the Athens metropolitan area (Figure 2a). The locations of the sampling sites are precisely presented in Figure 2b and Table 1. In considering the tropical origin [3,4,5,6] and the sensitivity to frost of *C. siliqua* [71], the sampling sites are cited according to increasing latitude (Table 1).

The even pinnate, compound leaves of *C. siliqua* consist of 6 or 8 leaflets, symmetrically arranged along the petiole; each leaflet (pinna) has a short petiolule (Figure 6). In this study, compound leaves of carob with 8 leaflets were used and nylon tags were applied on each compound leaf (Figure 6c). The opposite symmetrically arranged leaflets of *C. siliqua* possess very similar initiation and growth patterns [72,73].

The selected leaves were grown on the south-facing outer part of the canopy and exposed to day-long sunlight (Figure 6). Soon after the apex of the first leaf appeared, the top (apex) of the 15 shoots was labeled with colorful plastic cable ties. When nine compound leaves were developed (Figure 6c), the leaf collection was carried out. In each study site, three shoots of 9 successively grown sunny compound leaves (with 8 leaflets per compound leaf) along shoots were collected, early in the morning. Shoots from the meridian side of the trees (arrows in Figure 6a,b) were carefully selected and marked (see the shadow of the tree in Figure 6a). During the vegetative period the first leaf appearing on the young shoot was tagged. As the appearance of the new compound leaves was in progress, nylon tags were applied on each one of them until all the leaves on the stem were fully expanded. Among the tagged leaves, the apical one, which was the youngest among them, as well as the third, fifth and seventh in the row on the stem were used for the current investigation (Figure 6c). The edges of the cut shoots were wrapped with wet cotton, placed in sealed plastic bags and a portable refrigerator, and immediately transferred to the lab for analysis. 

### 4.2. Climate

The climate of the study sites was Mediterranean with a marked seasonality, typified by the alternation of a cold and wet period with a hot and dry period. Climatic data continuously collected by the National Weather Service of Greece (http://www.emy.gr, accessed on 1 December 2022) were obtained over the study period; the below mentioned temperature and precipitation values correspond to mean monthly data ± standard errors. Samples of 12-month-old, fully expanded leaves (i.e., developed during the previous year) were collected in late March of 2022. Samples of young expanding leaves of the current year, that is, 3-month-old expanding leaves were collected in early June 2022. The mean air temperature was 11.8 ± 0.3 °C in March and 25.8 ± 0.4 °C in June; the rainfall was 42 ± 3 mm in March and 7.2 ± 0.4 mm in June. The selected stands of trees were not irrigated throughout the year. 

### 4.3. Air Pollutants 

During the last two decades the measured concentrations of various air pollutants in the considered urban sites have been provided by the Greek Ministry of Environment and Energy (https://ypen.gov.gr, accessed on 9 January 2023); representative air pollutants measured throughout the year 2021–2022 in the research sites (KE, ME, KL, PF, NG) are presented in Table 5. 

### 4.4. Plant Anatomy and Microscopy

After each sampling, small pieces from leaflets at various developmental stages were fixed for the anatomical/microscopical investigation. The compound leaves were detached from the selected tree at each sampling site, and the third to the right leaflet, counting from the apex (i.e., the upper edge) of each compound leaf, was selected [73]. Numerous small pieces (1 × 1 mm) from the middle leaflet blades, adjacent to the central vein, were excised. All these pieces were fixed together in phosphate-buffered 3% glutaraldehyde (pH 6.8) at 0 °C for 2 h. The tissue was post-fixed in 1% osmium tetroxide in phosphate buffer, dehydrated in graded ethanol series, and embedded in Durcupan ACM (Fluka, Steinheim, Switzerland). Semithin sections, obtained from a LKB Ultrotome III (Sweden), were placed on glass slides and stained with 0.5 toluidine blue O (in 1% borax solution), as a general stain, for light microscopic observations. The protocols for double fixation, embedding, sectioning and light microscope observations have been published in detail [74]. Quantitative foliar anatomical and morphological traits were estimated on sections observed with light microscopy, that is, the thickness of leaflet lamina, palisade and spongy parenchyma, mesophyll, adaxial and abaxial epidermis, and adaxial and abaxial periclinal walls. The presented results are means of 30 measurements ± standard deviations.

### 4.5. Specific Leaf Area

Specific leaf area (SLA) was calculated by the ratio of the area of the compound leaves versus their dry mass (cm^2^ g^–1^) [75]; measurements were made on the same samples that were used for chlorophyll and phenolic content. Within half an hour after the harvest, five compound leaves successively expanded (i.e., the first, third, fifth and seventh leaf on axis, see Figure 6c) from the considered environmental sites were scanned in a flatbed scanner to calculate the fresh area (ImageJ Pro) and then oven-dried at 60 °C for 48 h to a constant mass and weighed to the nearest 0.001 g, and their dry weight was measured. Specific leaf area based on fresh leaf weight was found to suffer from a number of drawbacks, because it is both very variable among replicates and much influenced by experimental procedures, while leaf dry weight is constant, largely independent of the above-mentioned parameters [76,77]. The results are means of 30 measurements ± standard deviation.

### 4.6. Determination of Chlorophyll Content

The total chlorophyll (Chl) content was spectrophotometrically determined in dried leaf samples according to a modified acetone method [78]. The dried samples were powdered, using a MFC mill (Janke and Kunkel GMBH & Co, Staufen im Breisgau, Germany) and stored in tightly sealed containers, in a cool dry environment. Chlorophyll was extracted from 0.05 g dried, grounded leaf samples homogenized with 10 mL acetone (80% *v*/*v*) using porcelain pestle and mortar, and filtered through Whatman #2 filter paper to become fully transparent [79]. The chlorophyll content was measured in aliquots of the leaf extracts using a Novaspec II (Pharmacia, Biotech, Cambridge, UK) spectrophotometer; the absorbance readings of five replicates have been used for the calculations [17,80]. The values of total chlorophyll content (a + b) in the leaf tissue are expressed as mg g^−1^ of dry weight. The results are means of five replicates ± standard deviation.

### 4.7. Determination of Phenolic Content 

Total phenolic content in carob leaf extracts was determined by the Folin–Ciocalteu colorimetric method [81,82]. Briefly, 0.1 g of dry tissue, ground in an electric mill (Janke & Kunkel-Mikro-Feinmuhle-Cullati, IKA Labortechnik, Wasserburg, Germany), was immersed in 10 mL MeOH (50% *v*/*v*) and incubated in a water-bath for 3 h at 40 °C, while vortexed regularly. Phenolic compounds were extracted, filtered (Whatman #2 filter paper) and kept tightly sealed at 4 °C overnight. An aliquot (0.05 mL) of the diluted leaf extract [1:5 MeOH 10% (*v*/*v*)] was added to 3.95 mL of dH_2_O, 0.25 mL Folin–Ciocalteu reagent (that was previously diluted with water 1:10 *v*/*v*), 0.75 mL Na_2_CO_3_ 20% (*w*/*v*) and vortexed. The solution was maintained at 20 °C for 2 h and the absorption of the resulting colorimetric reaction was measured with a UV–VIS spectrophotometer (Pharmacia Biotech Novaspec II) at 760 nm. The total phenolic content was calculated using standard curves of tannic acid and expressed as mg of tannic acid equivalent per g (dry weight) of leaf tissues. For the calibration curve, 0.05 mL of the following tannic acid concentrations, that is, 0.02 mg/mL, 0.08 mg/mL, 0.16 mg/mL and 0.40 mg/mL, were added to 3.95 mL of dH_2_O, 0.25 mL Folin–Ciocalteu reagent (previously diluted with water 1:10 *v*/*v*), 0.75 mL Na_2_CO_3_ 20% (*w*/*v*), and vortexed. The absorbance of the resulting fluid of the colorimetric reaction (after 2 h at 20 °C) was measured with a UV–VIS spectrophotometer (Pharmacia Biotech Novaspec II) at 760 nm; the absorbance was plotted versus the concentration for the calibration curve. The results are means of five replicates ± standard deviation.

### 4.8. Statistical Analysis

Morphological measurements were performed independently on each individual leaf (*n* = 30). One-way analysis of variance (ANOVA test) was performed to compare the considered measurements among the sampling sites using IBM SPSS 20 (IBM, Armonk, NY, USA) Statistics. Statistical analysis was performed by employing paired *t*-tests comparing the samples collected from the different sites. No data transformation took place judging from the normality and homogeneity of variance tests. The differences were considered to be statistically significant when *p* < 0.05. Regression analysis was used to determine relationships among variables of expanding and expanded leaves. Principal Component Analysis (PCA), of the means of variables from carob’s foliar samples, was performed to detect any grouping of the specimens on the basis of the considered traits, using “ade4” and “adegraphics” packages [83,84]; the specific leaf area, thickness of lamina, thickness of palisade and spongy parenchyma, thickness of abaxial and adaxial periclinal walls, phenolic compounds and mesophyll thickness were taken into consideration. PCA was performed using the software OriginPro10 from Origin Labs. The data were centered in order to normalize the variables so that the considered traits were at the same scale to perform the visualization in a comprehensible way.

## 5. Conclusions

The study of the response of compound leaves of *C. siliqua* to urban and suburban sites provides some evidence for differences in sensitivity and exposure among its successively grown expanding and expanded leaves. Although there have been few detailed studies of this kind (in order to assess comparative results), it is likely that fully expanded leaves from urban sites exhibited an elevated ratio of palisade vs. spongy parenchyma and increased lamina thickness and chlorophyll content, but lower phenolic content. Moreover, the thickness of the adaxial and the abaxial epidermises, as well as phenolics and adaxial and abaxial periclinal walls were found to be elevated in suburban leaves. It is likely that expanding and expanded leaves of carob trees grown in urban sites have attained elevated SLA and chlorophyll content, respectively, probably counteracting the disadvantage of a greater uptake of pollutants per unit of SLA. 

## Figures and Tables

**Figure 1 plants-12-00514-f001:**
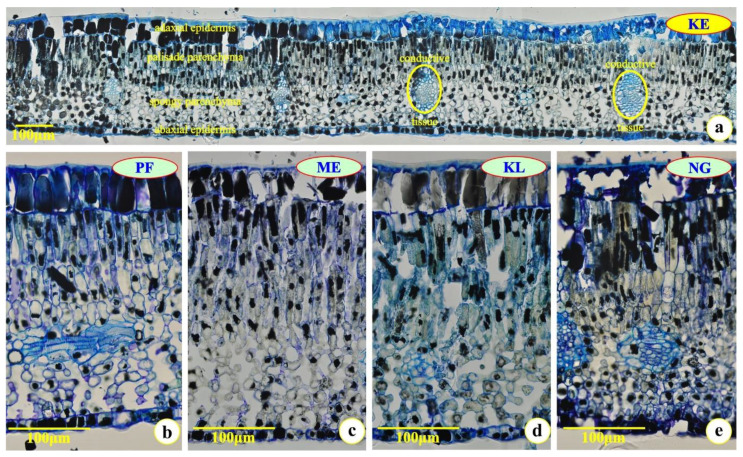
Cross-sectioned leaflets of fully expanded compound leaves from the selected trees. All sections were stained with toluidine blue “O”. (**a**) Leaf cross-section from the suburban site (KE), where the main histological features are indicated. (**b**) Leaf cross-section from the urban site PF, with the compact adaxial epidermis and the heavy load of phenolics within its cells. (**c**) Leaf cross-section from the urban site ME; the broken material out of and over the adaxial epidermis is condensed tannins. (**d**) Leaf cross-section from the urban site KL. (**e**) Leaf cross-section from the urban site NG that appears to have originated from a more xeromorphic plant as indicated by the heavier accumulation of phenolics within the mesophyll and the epidermal cells.

**Figure 2 plants-12-00514-f002:**
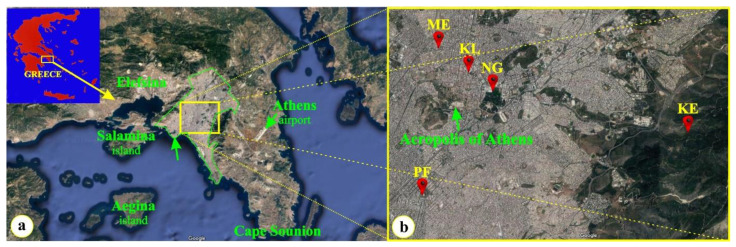
(**a**) Map of Attiki county and Athens metropolitan area, which is located southwest of the mainland of Greece (in the insert); the yellow square indicates the wider location of the sampling sites. (**b**) The five sampling sites indicated by yellow initials; four of them (PF, NG, KL and ME) are within the urban fabric, while KE is on the slope of Hymettus mountain.

**Figure 3 plants-12-00514-f003:**
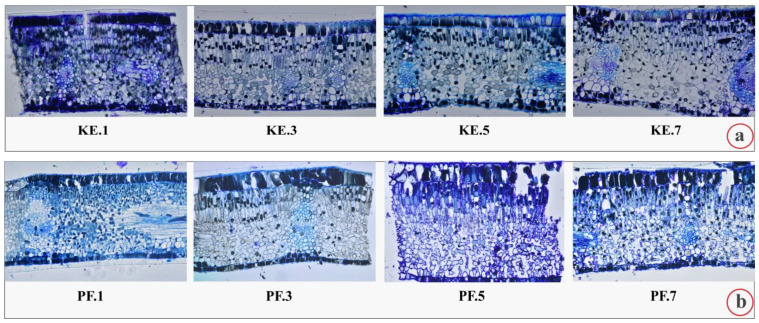
Cross-sections of the successively expanding leaflets of compound leaves of *C. siliqua* from selected trees and studied sites, that is, from: (**a**) KE (Kessariani aesthetic forest), (**b**), PF (Paleo Faliro), (**c**) KL (Klafthmonos square), (**d**) ME (Metaxourgeio) and (**e**) NG (National Garden); also, for capital initials see Table 1. KE.1, PF.1, KL.1, ME.1 and NG.1 correspond to cross-sections of the youngest first expanding leaflet. KE.3, PF.3, KL.3, ME.3 and NG.3 correspond to cross-sections of the third expanding leaflet. KE.5, PF.5, KL.5, ME.5 and NG.5 correspond to cross-sections of the fifth expanding leaflet. KE.7, PF.7, KL.7, ME.7 and NG.7 correspond to cross-sections of the seventh expanding leaflet (see Section 4.1). The adaxial epidermis is on the upper part of each image. All sections were stained with toluidine blue “O” and are presented under the same magnification.

**Figure 4 plants-12-00514-f004:**
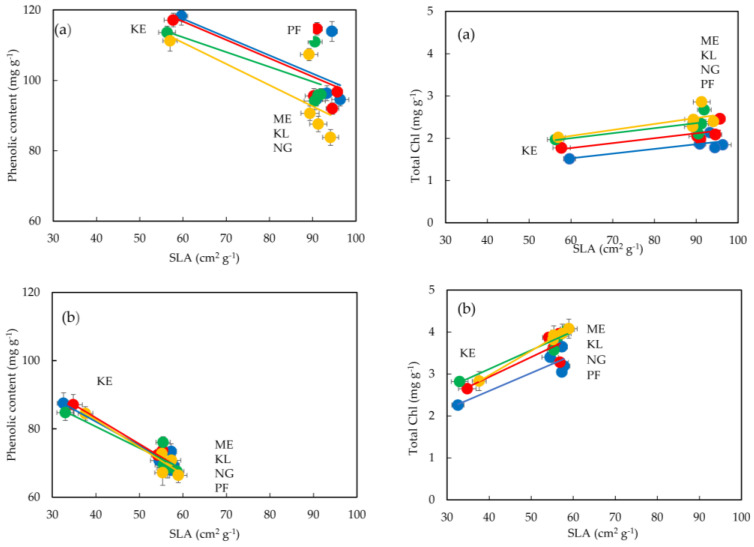
Linear correlation of specific leaf area (SLA) versus phenolic (**left**) and chlorophyll (**right**) content in expanding (**a**) and expanded (**b**) compound leaves of *C. siliqua*; blue circles indicate the first leaf, red circles indicate the third leaf, green circles indicate the fifth leaf, and yellow circles indicate the seventh leaf. The five sampling sites are indicated by initials; four of them (PF, NG, KL and ME) are within the urban fabric, while KE is on the suburban site on Hymettus Mountain. The results are means (see Section 4.5, Section 4.6 and Section 4.7) ± standard deviation.

**Figure 5 plants-12-00514-f005:**
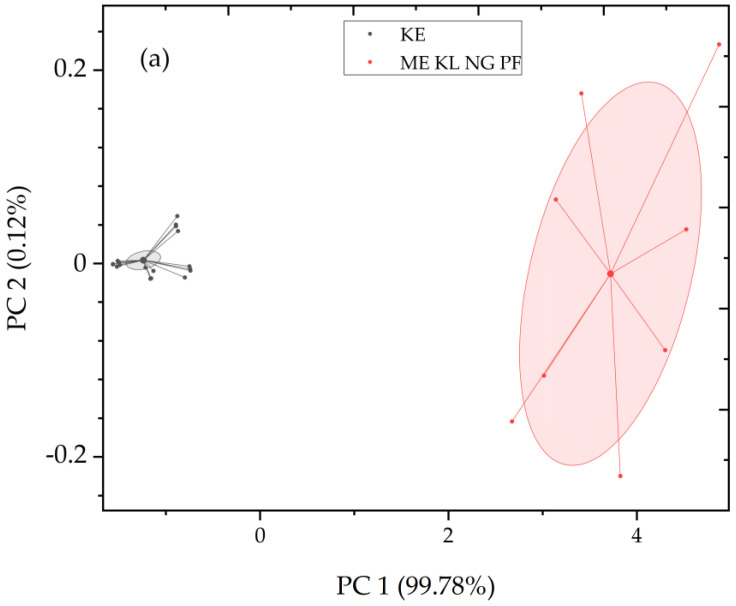
(**a**) Visualization of PCA based on the considered morphological and physiological traits of successively grown expanded carob leaves from suburban (KE) and urban (ME, KL, NG, PF) sites, revealing the substantial grouping of compound leaves of *C. siliqua* derived from urban polluted sites (pink) versus suburban (grey) habitat. (**b**) PCA based on traits of carob leaflets from urban (ME, KL, NG, PF) and suburban (KE) sites depicted using red spots and letters; on the black axes, the principal components are presented, and via the blue axes, the contribution of each variable on the principal components. Blue lines assigned to the variables, represent principal component loadings associated with individual traits; at the edges of the blue lines, the numbers 1, 3, 5 and 7 (in blue) correspond to the foliar developmental stage, and the letters (in blue) indicate the considered variables, that is, the specific leaf area (SLA), thickness of lamina (TLL), thickness of palisade (PS) and spongy (SP) parenchyma, thickness of abaxial (ABP) and adaxial (ADP) periclinal walls, phenolic compounds (PHE) and mesophyll thickness (INT).

**Figure 6 plants-12-00514-f006:**
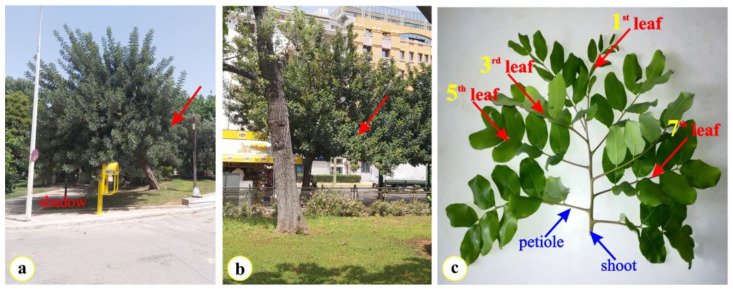
Two, among the five selected trees for sampling, are depicted. (**a**) The carob tree grown in the site Paleo Faliro (PF) and (**b**) in front of the National Garden (NG). The red arrows in (**a**) and (**b**) indicate the meridian site of the carob tree. (**c**) A young shoot from a carob tree; the considered successive compound leaves of *C. siliqua* used in this investigation are pointed out using arrows.

**Table 1 plants-12-00514-t001:** The sampling sites cited according to the declined succession of their latitude.

Names of Sampling Sites (Initials)	Latitude	Longitude
Metaxourgeio (ME)	37.985817 E	23.721294 N
Klafthmonos sq (KL)	37.979542 E	23.731402 N
National Garden (NG)	37.975728 E	23.738600 N
Kessariani aesthetic forest (KE)	37.962400 E	23.796639 N
Paleo Faliro (PF)	37.930629 E	23.717089 N

**Table 2 plants-12-00514-t002:** Thickness of morphological traits of expanding and expanded leaflets of the compound leaves of *C. siliqua*. The values are means of 30 replicates ± standard deviation. Different lower-case, superscript letters next to each value correspond to statistically significant differences of each parameter in each developmental stage of the leaflets between sites (one-way ANOVA, *t*-test at *p* < 0.05; for all cases *n* = 30).

Sites	Leaflet (μm)	Periclinal Wall of Adaxial Epidermal Cells (μm)	Adaxial Epidermis (μm)	Mesophyll (μm)	Palisade Parenchyma (μm)	Spongy Parenchyma (μm)	Abaxial Epidermis (μm)	Periclinal Wall of Abaxial Epidermal Cells (μm)
				**first young**	**expanding**			
ME	702.00 ± 12.35 ^b^	8.31 ± 0.19 ^c^	102.68 ± 8.97 ^a^	556.91 ± 8.53 ^a^	240.18 ± 9.36 ^ab^	316.73 ± 8.25 ^ab^	39.95 ± 2.98 ^c^	3.99 ± 0.99 ^c^
KL	711.81 ± 12.86 ^a^	8.98 ± 1.38 ^b^	105.01 ± 4.89 ^a^	558.93 ± 8.38 ^a^	238.23 ± 7.56 ^b^	320.70 ± 9.36 ^a^	45.92 ± 2.97 ^b^	5.66 ± 0.81 ^b^
NG	579.84 ± 9.67 ^d^	7.44 ± 0.90 ^d^	69.55 ± 2.54 ^c^	466.07 ± 9.74 ^c^	197.02 ± 7.48 ^c^	269.06 ± 10.20 ^d^	41.62 ± 4.69 ^bc^	4.51 ± 1.22 ^c^
KE	667.90 ± 10.97 ^c^	10.59 ± 0.77 ^a^	95.56 ± 5.85 ^b^	527.43 ± 9.56 ^b^	246.94 ± 8.27 ^a^	300.49 ± 9.82 ^c^	48.98 ± 3.07 ^a^	9.97 ± 1.57 ^a^
PF	712.94 ± 10.36 ^a^	8.15 ± 1.27 ^c^	100.59 ± 5.36 ^a^	559.39 ± 8.29 ^a^	244.91 ± 9.33 ^ab^	314.47 ± 10.29 ^ab^	49.73 ± 3.80 ^a^	5.47 ± 1.26 ^b^
**third expanding**								
ME	759.51 ± 7.49 ^a^	9.94 ± 1.46 ^b^	120.43 ± 3.53 ^a^	579.52 ± 8.24 ^a^	269.25 ± 9.70 ^a^	310.28 ± 9.45 ^a^	53.49 ± 3.36 ^b^	5.33 ± 1.13 ^c^
KL	758.49 ± 1.99 ^a^	9.55 ± 2.61 ^b^	120.02 ± 9.14 ^a^	582.73 ± 9.23 ^a^	267.88 ± 8.30 ^a^	314.86 ± 8.56 ^a^	53.61 ± 3.69 ^b^	5.98 ± 1.55 ^c^
NG	668.40 ± 6.93 ^d^	8.82 ± 1.44 ^c^	95.04 ± 3.39 ^c^	530.75 ± 9.94 ^c^	240.23 ± 7.88 ^c^	290.52 ± 9.46 ^c^	42.57 ± 3.65 ^c^	6.82 ± 1.10 ^b^
KE	716.21 ± 7.49 ^c^	12.00 ± 1.46 ^a^	106.34 ± 3.53 ^b^	560.86 ± 8.24 ^b^	258.43 ± 9.70 ^b^	302.43 ± 9.45 ^b^	55.08 ± 3.36 ^a^	10.01 ± 1.13 ^a^
PF	744.19 ± 10.36 ^b^	9.31 ± 0.96 ^b^	115.74 ± 2.54 ^a^	575.51 ± 9.87 ^a^	254.22 ± 10.22 ^b^	311.29 ± 8.25 ^a^	51.04 ± 2.45 ^b^	5.86 ± 1.02 ^c^
**fifth expanding**								
ME	760.77 ± 9.94 ^b^	12.87 ± 0.77 ^a^	122.59 ± 4.03 ^b^	591.76 ± 4.88 ^a^	272.47 ± 11.01 ^a^	319.29 ± 11.57 ^b^	52.85 ± 3.74 ^b^	7.64 ± 0.42 ^b^
KL	762.49 ± 8.71 ^b^	12.99 ± 0.76 ^a^	123.06 ± 5.93 ^b^	583.91 ± 10.69 ^b^	270.80 ± 7.45 ^a^	313.11 ± 10.73 ^b^	52.67 ± 4.36 ^b^	7.19 ± 0.87 ^b^
NG	723.57 ± 8.96 ^d^	10.27 ± 0.32 ^b^	111.75 ± 7.54 ^c^	563.69 ± 12.45 ^c^	267.11 ± 10.23 ^b^	296.57 ± 10.14 ^c^	43.37 ± 4.28 ^c^	7.55 ± 0.06 ^b^
KE	768.34 ± 6.90 ^c^	12.70 ± 0.26 ^a^	115.78 ± 8.77 ^c^	600.29 ± 7.02 ^a^	272.91 ± 5.89 ^a^	327.38 ± 1.98 ^a^	55.23 ± 5.89 ^a^	11.85 ± 0.11 ^a^
PF	784.15 ± 9.56 ^a^	10.09 ± 0.29 ^b^	138.58 ± 8.26 ^a^	597.16 ± 13.78 ^a^	268.55 ± 9.88 ^b^	328.61 ± 10.27 ^a^	51.16 ± 3.98 ^b^	6.75 ± 0.21 ^c^
**seventh expanding**								
ME	808.52 ± 8.89 ^a^	13.10 ± 1.55 ^b^	124.52 ± 6.74 ^b^	637.85 ± 12.19 ^a^	291.62 ± 7.16 ^a^	346.23 ± 15.16 ^a^	53.26 ± 1.99 ^b^	8.83 ± 1.52 ^c^
KL	806.83 ± 7.67 ^a^	13.02 ± 1.65 ^b^	122.01 ± 5.08 ^b^	634.60 ± 12.83 ^a^	290.74 ± 7.57 ^a^	343.86 ± 10.11 ^a^	53.57 ± 2.01 ^b^	8.66 ± 1.64 ^c^
NG	808.43 ± 9.35 ^a^	11.33 ± 1.47 ^c^	120.38 ± 5.02 ^b^	631.43 ± 10.32 ^a^	290.77 ± 6.99 ^a^	340.66 ± 10.78 ^a^	49.74 ± 2.77 ^c^	8.80 ± 1.06 ^c^
KE	787.03 ± 7.37 ^b^	14.26 ± 1.26 ^a^	123.06 ± 5.37 ^b^	609.55 ± 1.98 ^b^	275.17 ± 1.32 ^c^	338.80 ± 2.72 ^b^	57.05 ± 0.49 ^a^	12.01 ± 0.12 ^a^
PF	804.86 ± 9.86 ^a^	11.85 ± 2.89 ^c^	139.78 ± 7.02 ^a^	600.05 ± 10.64 ^c^	280.26 ± 10.65 ^b^	319.79 ± 10.03 ^c^	58.81 ± 5.33 ^a^	10.13 ± 2.45 ^b^
**third expanded**								
ME	951.00 ± 9.26 ^b^	13.98 ± 0.63 ^c^	133.95 ± 2.56 ^c^	757.23 ± 6.27 ^c^	438.41 ± 5.87 ^b^	318.82 ± 7.96 ^c^	57.36 ± 2.90 ^c^	9.22 ± 0.68 ^c^
KL	959.63 ± 7.70 ^b^	13.57 ± 0.40 ^c^	135.01 ± 5.22 ^c^	762.15 ± 6.42 ^c^	443.01 ± 5.49 ^b^	319.14 ± 6.84 ^c^	57.93 ± 2.57 ^c^	9.35 ± 0.49 ^c^
NG	1086.23 ± 5.31 ^a^	12.89 ± 0.45 ^d^	134.05 ± 3.49 ^c^	816.63 ± 4.73 ^a^	467.90 ± 5.54 ^a^	308.73 ± 6.27 ^d^	56.94 ± 1.03 c	9.12 ± 0.46 ^c^
KE	838.00 ± 4.36 ^d^	16.65 ± 0.64 ^a^	157.00 ± 3.53 ^a^	623.45 ± 9.22 ^d^	289.36 ± 7.25 ^d^	334.09 ± 6.51 ^b^	70.24 ± 2.86 ^a^	13.02 ± 0.88 ^a^
PF	879.56 ± 8.15 ^c^	14.96 ± 0.60 ^b^	142.35 ± 4.36 ^b^	714.96 ± 4.60 ^b^	325.23 ± 8.11 ^c^	349.73 ± 6.32 ^a^	60.87 ± 2.42 ^b^	11.02 ± 0.89 ^b^

**Table 3 plants-12-00514-t003:** Specific leaf area (SLA) of expanding and expanded compound leaves of *C. siliqua*; the values are means of 30 replicates ± standard deviation. Different lower-case, superscript letters next to each value correspond to statistically significant differences of SLA in each developmental stage, that is, expanding and expanded leaflets in each research site (one-way ANOVA, *t*-test at *p* < 0.05; for all cases *n* = 30).

SLA (cm^2^g^−1^)	KE	ME	KL	NG	PF
**expanding leaf**					
first	54.64 ± 0.59 ^b^	90.87 ± 0.82 ^a^	93.36 ± 0.30 ^a^	91.29 ± 0.48 ^a^	90.47 ± 0.18 ^a^
third	57.74 ± 0.66 ^a^	90.26 ± 0.22 ^a^	94.58 ± 0.21 ^a^	92.69 ± 0.26 ^a^	90.98 ± 0.22 ^a^
fifth	57.31 ± 0.91 ^a^	90.45 ± 0.45 ^a^	94.22 ± 0.22 ^a^	91.93 ± 0.55 ^a^	90.54 ± 0.32 ^a^
seventh	57.01 ± 0.35 ^a^	89.36 ± 0.34 ^a^	94.13 ± 0.89 ^a^	91.30 ± 0.43 ^a^	89.16 ± 0.67 ^a^
**expanded leaf**					
first	33.56 ± 0.42 ^b^	57.82 ± 0.61 ^a^	54.54 ± 0.74 ^b^	56.38 ± 0.44 ^b^	56.28 ± 0.78 ^a^
third	34.79 ± 0.47 ^a^	56.86 ± 0.96 ^a^	54.16 ± 0.92 ^b^	56.68 ± 0.15 ^b^	55.31 ± 0.72 ^b^
fifth	34.97 ± 0.18 ^a^	56.47 ± 0.29 ^a^	55.57 ± 0.97 ^a^	58.59 ± 0.74 ^a^	55.44 ± 0.40 ^b^
seventh	35.61 ± 0.44 ^a^	57.37 ± 0.32 ^a^	55.35 ± 0.71 ^a^	58.93 ± 0.79 ^a^	55.18 ± 0.46 ^b^

**Table 4 plants-12-00514-t004:** Linear regressions of total chlorophyll and phenolic content versus specific leaf area (SLA) of expanding and fully expanded leaves of *C. siliqua*.

**Expanding leaf**	**Linear regression of Chl *versus* SLA**
first	y = 0.0112x + 0.8681, *r* = 0.5924, *p* < 0.05
third	y = 0.0116x + 1.0718, *r* = 0.5559, *p* < 0.05
fifth	y = 0.0117x + 1.2993, *r* = 0.4814, *p* < 0.05
seventh	y = 0.0141x + 1.2137, *r* = 0.5014, *p* < 0.05
**Expanded leaf**	
first	y = 0.0286x + 0.5587, *r* = 0.9525, *p* < 0.05
third	y = 0.0249x + 1.2466, *r* = 0.9873, *p* < 0.05
fifth	y = 0.0314x + 1.0531, *r* = 0.9994, *p* < 0.05
seventh	y = 0.0317x + 1.0323, *r* = 0.9915, *p* < 0.05
**Expanding leaf**	**Linear regression of Phenolics *versus* SLA**
first	y = −0.5182x + 148.56, *r* = 0.4714, *p* < 0.05
third	y = −0.5241x + 148.19, *r* = 0.5013, *p* < 0.05
fifth	y = −0.4214x + 137.52, *r* = 0.4862, *p* < 0.05
seventh	y = −0.6034x + 146.92, *r* = 0.5609, *p* < 0.05
**Expanded leaf**	
first	y = −0.7038x + 110.32, *r* = 0.9324, *p* < 0.05
third	y = −0.7659x + 113.86, *r* = 0.9663, *p* < 0.05
fifth	y = −0.6260x + 105.72, *r* = 0.8408, *p* < 0.05
seventh	y = −0.7979x + 114.55, *r* = 0.9027, *p* < 0.05

**Table 5 plants-12-00514-t005:** Mean NO, NO_2_ and O_3_ (µg m^−3^) values at the considered sites; there were no data recorded for the missing values.

Time	KE	KE	KE	ME	ME	ME	KL	KL	KL	PF	PF	PF	NG	NG	NG
Month, Year	NO	NO_2_	O_3_	NO	NO_2_	O_3_	NO	NO_2_	O_3_	NO	NO_2_	O_3_	NO	NO_2_	O_3_
January 2022	2	16	64	89	44	9	48	30	15	31	40	59	49	28	16
January 2021	2	9	50	52	41	8	23	19	25	15	25	55	22	20	24
February 2022	2	15	70	55	46	11	32	26	22	18	38	-	31	27	23
February 2021	2	11	57	60	48	13	29	23	22	19	35	30	30	23	21
March 2022	1	12	72	37	40	24	18	20	30	5	25	-	19	21	31
March 2021	1	12	66	19	39	19	16	25	43	6	26	60	17	25	42
April 2022	1	7	86	14	33	40	8	14	44	2	9	-	9	14	40
April 2021	2	12	66	18	42	19	18	28	60	5	25	81	12	28	58
May 2022	1	10	84	25	43	19	11	15	21	-	-	-	12	14	20
May 2021	1	7	88	15	48	12	13	30	62	5	30	83	12	29	53
June 2022	2	11	81	15	37	29	9	13	11	-	-	-	10	14	13
June 2021	1	11	105	15	47	15	15	31	60	5	29	88	16	30	59
July 2021	1	6	115	10	46	41	17	35	67	5	20	95	17	36	65
August 2021	1	8	113	15	43	33	25	35	50	-	26	90	24	35	48
September 2021	1	8	9 6	29	53	30	33	45	47	-	25	80	32	44	47
October 2021	1	7	71	30	45	26	28	32	38	-	23	68	28	30	36
November 2021	2	10	58	56	48	14	38	34	32	13	30	54	37	33	30
December 2021	2	10	57	67	33	11	43	36	33	12	29	48	39	32	33

## Data Availability

The data are available from the authors upon request.

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
