# Peer review of "Structural and Physiological Traits of Compound Leaves of *Ceratonia siliqua* Trees Grown in Urban and Suburban Ambient Conditions"

_plants, 2023, doi:10.3390/plants12030514_

Round 1
Reviewer 1 Report
The manuscript “Structural and physiological traits of compound leaves of Ceratonia siliqua trees grown in urban and suburban ambient conditions” describes the response of compound leaves of C. siliqua to urban and suburban sites. This is a study on a topic of high significance. However, the manuscript needs a minor revision, following the points listed below, to be fully published.
1) The abstract should be rewritten because there is confusion about the order of the parts that commonly characterize this section of the manuscript. In fact, an abstract should be summarized in the following order: 1) Background; 2) Methods; 3) Results; 4) Conclusion.
2) The quality of the images and graphs is very low. Please, improve them. Usually, the standard quality of an image is 300 DPI.
3) Line 262: Please check the word “periole”.
4) Line 280: Please define the type of transponders (tags) that you used.
5) Lines 301-308: Please add a definition about the symbol ± (e.g. standard error, standard deviation etc.)
Author Response
the response is attached

Reviewer 2 Report
The paper reports the response of Ceratonia siliqua to air pollutants.
The manuscript does not meet the standard quality of Plants, therefore, my recommendation is major revision.
Introduction:
The introduction section should be improved to better establish the main points of the manuscript
Materials and method:
In the introduction, it is stated that the leaves ceased their growth after 3 months. Why in this experiment leaves of 4 months are considered expanding?
I recommend the authors to remove from line 263 to 271. That information does not add any value to the manuscript
The section 4.3 Air pollutants could improve if the data were presented in a table.
In the section 4.4 Plant anatomy and microscopy. At the end of the paragraph “The morphological traits were estimated on sections observed with light microscopy; several anatomical features were measured.” Describe which features were measured.
The quality of the figures is low. For example, in the leaf histology figures it is not possible to determine anything in them
Statistical analysis:
The authors affirmed that they perform Anovas, however, no information along the text were offered to their results.
Why the authors did not perform a post-hoc analysis to determine the differences between zones in case the Anova was significant?
No comment is offered as to whether the authors checked the assumptions of the anovas. This is important to interpret correctly the data.
Which variables were used for the PCA analysis?
I recommend the authors to perform a PCA biplot to analyze in the same figure the treatments and the variables
Discussion:
This section is too descriptive, failing to address the reasons behind some of the findings.
Line 213-215. In the reference 50 the chlorophyll concentration decreased in the most contaminated zone, so it is the opposite. And in the reference 51 if the contamination is made of ozone the chlorophyll concentration decreased while increased if the pollutant is NO. Please, reflect that.
Conclusions:
English language and style should be improved. For example:
Abstract line 1-3: The verb is missing.
Abstract line 5: Remove the comma after While
Line 262: symmetrically is misspelled
Line 294: Remove the comma before were
Minor comments:
Line 326. It should be cm2 g-1
In the section 4.6 and 4.7 standard deviation is abbreviated and, in some cases, also dry weight. However, in the rest of the manuscript they are not abbreviated. Please, be consistent
Author Response
the response is attached

Round 2
Reviewer 2 Report
The manuscript is improved and my recommendation is accept in the present form